# Effect on Body Weight and Adipose Tissue by Cariprazine: A Head-to-Head Comparison Study to Olanzapine and Aripiprazole in Rats

**László-István Bába** [1], **Zsolt Gáll** [1], **Melinda Kolcsár** [1,*], **Zsuzsánna Pap** [2], **Zoltán V. Varga** [3], **Béla Kovács** [4], **Beatrix Hack** [5] and **Imre-Zoltán Kun** [6]

[1] Department of Pharmacology and Clinical Pharmacy, George Emil Palade University of Medicine, Pharmacy, Science and Technology of Targu Mures, 540142 Targu Mures, Romania; laszlo.baba@umfst.ro (L.-I.B.); zsolt.gall@umfst.ro (Z.G.)

[2] Department of Anatomy, George Emil Palade University of Medicine, Pharmacy, Science and Technology of Targu Mures, 540142 Targu Mures, Romania; zsuzsanna.pap@umfst.ro

[3] Department of Pharmacology and Pharmacotherapy, Semmelweis University, 1085 Budapest, Hungary; varga.zoltan@med.semmelweis-univ.hu

[4] Department of Pharmaceutical Biochemistry and the Chemistry of Environmental Factors, George Emil Palade University of Medicine, Pharmacy, Science and Technology of Targu Mures, 540142 Targu Mures, Romania; bela.kovacs@umfst.ro

[5] Faculty of Medicine, George Emil Palade University of Medicine, Pharmacy, Science and Technology of Targu Mures, 540142 Targu Mures, Romania; beatrixhack@gmail.com

[6] Doctoral School, George Emil Palade University of Medicine, Pharmacy, Science and Technology of Targu Mures, 540142 Targu Mures, Romania; kunimre@gmail.com

* Correspondence: melinda.kolcsar@umfst.ro; Tel.: +40-748-836-224

**Abstract:** Cariprazine (Car) is a recently approved second generation antipsychotic (SGA) with unique pharmacodynamic profile, being a partial agonist at both dopamine $D_{2/3}$ receptor subtypes, with almost 10 times greater affinity towards $D_3$. SGAs are known to increase body weight, alter serum lipids, and stimulate adipogenesis but so far, limited information about the adverse effects is available with this drug. In order to study this new SGA with such a unique mechanism of action, we compared Car to substances that are considered references and are well characterized: olanzapine (Ola) and aripiprazole (Ari). We studied the effects on body weight and also assessed the adipogenesis in rats. The drugs were self-administered in two different doses to female, adult, Wistar rats for six weeks. Weekly body weight change, vacuole size of adipocytes, Sterol Regulatory Element Binding Protein-1 (SREBP-1) and Uncoupling Protein-1 (UCP-1) expression were measured from the visceral adipose tissue (AT). The adipocyte's vacuole size, and UCP-1 expression were increased while body weight gain was diminished by Car. by increasing UCP-1 might stimulate the thermogenesis, that could potentially explain the weight gain lowering effect through enhanced lipolysis.

**Keywords:** second generation antipsychotic; metabolic syndrome; cariprazine; aripiprazole; olanzapine; adipose tissue

## 1. Introduction

Second generation antipsychotics (SGAs) are drugs used in the treatment of severe mental health issues, including schizophrenia. Their adverse effects differ from those of the first generation and comprise mainly metabolic derangements [1]. The severity of these varies notably. The relative risk of weight gain decreases in the following order: clozapine (Clo) ≥ olanzapine (Ola) > quetiapine > risperidone > sertindole > zotepine > aripiprazole (Ari) [1–3].

Car recently received FDA approval for the treatment of schizophrenia and manic/mixed episodes associated with type one bipolar disorder [4]. At this moment, there are limited data in the literature regarding the mAE of Car. Mild weight gain and blood-lipid lowering effects were reported so far [4–6]. A potential aspect, that might help to understand the mAE of SGA is the receptor-binding profile of a drug. As reviewed by Nasrallah, there is a close relationship between affinity towards $5\text{-HT}_{2C}$, $5\text{-HT}_{1A}$ and $H_1$ receptors and the mAE of SGA [7]. In this regard, Car has a favorable profile (similar to that of Ari), having high affinity only for the $5\text{-HT}_{1A}$ receptor with partial agonist activity, and weak affinity towards $5\text{-HT}_{2C}$ and $5\text{-HT}_{2A}$ (again similar to Ari) [6]. The mAE of Ola are very pronounced and accordingly, the drug has medium affinity towards $H_1$ receptor and higher $5\text{-HT}_{2C}$ receptor affinity than Ari and Car [4,7]. In order to characterize Car, we compared it with Ola and Ari in a head-to-head design experiment mimicking chronic SGA administration. While Ola is situated on one extreme of the scale (having very pronounced mAE), Ari represents the opposite extreme on the scale being one of the safest agents in this perspective (known to be weight neutral) [1–3]. One aim of this study was to place Car on this virtual scale by observing the body weight change.

AT is an organ with a well-defined role in energy metabolism, like in the mAE of SGAs [8]. It is worth mentioning that, although there is a certain correlation between body mass index and metabolic diseases, metabolic derangements in obesity display a tighter relation with pathological/ectopic (visceral/abdominal) AT mass called "adiposopathy" (i.e., "sick fat") [9]. In this concern, the ability of a drug to increase the visceral fat mass can be a key element in the chain of events leading to metabolic syndrome. Clearly, alterations of the function of this organ can lead to metabolic diseases, thus its role in the pathogenesis of metabolic alterations is permanently emerging [9]. We decided to examine the perirenal AT for the assessment of AT-related effects, because this type of visceral AT expresses UCP-1, and also β3 adrenoreceptors and has considerable metabolic flexibility (i.e., capacity to accumulate TG to store energy and also ability for on-demand browning) [10].

Adipogenesis is a complex phenomenon orchestrated by a range of regulators, including transcription factors, micro-RNAs, hormones, adipokines [11]. As a result of this process, preadipocytes develop a distinct phenotype characterized morphologically by a large lipid vacuole. To assess the net effect of SGA treatment on adipogenesis, the vacuole size of the visceral AT adipocytes was measured after treatment with the mentioned SGAs.

A transcription factor with central role in adipogenesis is Sterol Regulatory Element Binding Protein-1 (SREBP-1). Transcription of this factor at the adipocyte level is controlled, among others, by insulin [12]. In this cell type, most of the insulin action are mediated by SREBP-1. As a promoter of adipogenesis, SREBP-1 is critical in adipocyte development [13,14]. Important downstream events to SREBP-1 activation are induction of peroxisome proliferator-activated receptor γ (PPARγ) mRNA and protein synthesis, and also the production of endogenous PPAR ligands [15,16]. Furthermore, the expression of SREBP-1 increases the activity of PPARγ [16]. Therefore, the metabolic profile of SGAs can be influenced by the induction/suppression of SREBP-1. This phenomenon has already been described in the case of Ola, clozapine and risperidone in various experimental settings [17–19]. The effect of Ari on this factor is not clear (some authors reported a slight, insignificant decrease in SREBP-1 expression in parametrial AT) [18]. To the best of our knowledge, there are no data regarding the expression of this factor after Car treatment.

A transporter with critical role in energy metabolism is uncoupling protein-1 (UCP-1, also called thermogenin). UCP-1 is responsible for the transport of protons through the internal membrane of the mitochondria resulting eventually in heat, instead of ATP [20]. After the discovery of brown AT in the adult human, the biology and function of brown AT gained enormous scientific interest [21]. A phenomenon called browning (i.e., transdifferentiation of white AT resulting in a phenotype with considerable UCP-1 expression, called beige AT) has also been described. The effect of SGA on brown AT and the browning process might be a possible explanation for the mAE of the SGA. Ola and clozapine were previously shown to decrease the expression of UCP-1, in vitro and in vivo experimental settings [10,22,23]. This represents a mechanism that lowers the energy expenditure through lowering

the thermogenesis. No data are currently available for Car regarding the influence on the expression of this transporter. By measuring SREBP-1 and UCP-1 expression and vacuole size quantification, we intended to contribute to the elucidation of AT-related effects of the three mentioned drugs.

## 2. Materials and Methods

### 2.1. Drugs

Car (99.53% purity) was obtained from Sanghai Resuperpharm Tech Ltd., Shanghai, China; in the case of Ola and Ari, commercially available products were used, freshly powdered and utilized for the preparation of cookie pellets (for Ola–Olanzapin, oral rapidly dissolving 20 mg tablets were purchased—Actavis, for Ari–Aryzalera, coated tablets containing 10 mg of Ari-Krka, as available for human use).

### 2.2. Materials for Immunohistochemistry

Primary and secondary antibody, as well as the blocking agent were acquired from ABCAM plc., Cambridge, England (rabbit polyclonal anti-UCP-1 antibody cat. nr.: ab23841, rabbit polyclonal anti-SREBP-1 cat. nr.: ab28481, goat Anti-Rabbit IgG H&L (HRP) nr.: ab205718, normal goat serum: nr. ab7481).

### 2.3. Animals and Study Design

In order to mirror the chronic administration of SGAs to human subjects, Ola, Ari, Car were administered orally, in two different doses to adult (at least 20 weeks old, at the start of the study), female Wistar rats ($n = 6$ each group, a total of $6 \times 7 = 42$) for six weeks (Table 1).

**Table 1.** The treated/control groups, and dosage.

| Group | Dosage [a] |
|---|---|
| Ctr | Vehicle |
| Ola 1.5 mg/kg | $3 \times 0.5$ |
| Ola 6 mg/kg | $3 \times 2$ |
| Ari 1.5 mg/kg | $3 \times 0.5$ |
| Ari 3 g/kg | $3 \times 1$ |
| Car 0.1 mg/kg | $1 \times 0.1$ |
| Car 0.25 mg/kg | $1 \times 0.25$ |

[a] Doses are expressed as mg/kg/day.

The rats were obtained from the Biobasis of the G. E. Palade University of Medicine, Pharmacy, Science and Technology of Tg. Mures, and were housed separately throughout the whole experimental procedure. We decided to use female animals (without synchronizing their estrous cycle), because in male rats, the mAE of SGAs cannot be robustly reproduced [22]. Doses were chosen considering earlier experiments in such a manner, that they mimicked the doses utilized in human therapy [24–26]. One group served as control (Ctr), these animals were administered pellets containing only vehicle. The drugs were self-administered orally, as previously described by Weston-Green et al. in dry-dough cookie pellets [24]. First, animals were left for acclimation for 10 days; meanwhile, they were conditioned to consume the pellets (administration of vehicle pellets); then, the administration of drug-containing pellets was initiated. Chronic administration can be considered a period longer than 3 weeks, therefore the duration of treatment period (six weeks) was chosen based on earlier literature data [27,28]. During the administration period, the animals were observed to be sure they all consumed the pellets. In this manner, the stress of immobilization and/or parenteral administration was eliminated. Ad libitum access to a standard laboratory chow diet and tap water was ensured during the whole experiment. Body weight was weekly measured in order to observe the treatment's effect

and also for the dose adjustments. After four weeks of treatment, one animal from the Ola 6 mg/kg group discontinued the consumption of the drug-containing pellets and was sacrificed; thus, the results for this group were calculated from the remaining five animals. At the end of the six-weeks treatment period, the animals were sacrificed by overdose of dissociative anesthetic in combination with $\alpha_2$ adrenergic receptor agonist followed by exsanguination by cardiac puncture (ketamine-xylazine: 300 and 30 mg/kgbw, respectively), as recommended by the American Veterinary Medical Association guidelines for the euthanasia of animals (2020 edition) [29]. Ketamine was obtained from Bela-Pharm Gmbh & Co. KG, Vechta, Germany, and xylazine from Bioveta a. s., Ivanovice na Hane, Czech Republic). Perirenal AT specimens were collected, and kept in alcohol until paraffin embedding.

### 2.4. Ethic Statement

All experimental procedures were carried out in accordance with EU guidelines (Directive 2010/63/UE of the European Parliament and of the European Council for the protection of animals used for scientific purposes) for experimental use of animals, and also received the approval of the Ethics Committee for Scientific Research of the George Emil Palade University of Medicine, Pharmacy, Science, and Technology of Targu Mures (approval no. 276/10.08.2017).

### 2.5. Immunohistochemistry

SREBP-1 and UCP-1 protein expression (reflected by the immunoreactivity per section) in AT were assessed by immunohistochemistry. The AT specimens were paraffin embedded, and five-micrometer-thick tissue sections were cut (one from each animal). After deparaffinization, the endogenous peroxidase was blocked by a 10-min 3% $H_2O_2$ bath followed by heat-induced epitope retrieval with pH = 10 and pH = 6 solutions for SREBP-1 and UCP-1 determination, respectively. According to the manufacturer's recommendations, non-specific epitopes were blocked for two hours with 2.5% normal goat serum in PBS. Incubation with primary antibody was carried out at 4 °C overnight. The applied dilutions were 1:400 for SREBP-1 and 1:500 for UCP-1. Secondary antibody was applied in 1:5000 dilution for one hour. 3,3'-diaminobenzidine (DAB) chromogen was used for detecting the primary antibodies. This was followed by Hematoxylin staining. For the positive control, liver and interscapular brown AT sections were processed in parallel, while for the estimation of nonspecific binding of the secondary antibody, a determination with the primary antibody omitted was performed as recommended by Hewitt and coworkers [30]. Images of sections were taken using a Nikon Eclipse e600 optical microscope at 200× magnification. The images were compared in terms of positive reaction in ImageJ software (R: 40–120, G: 20–60, B: 20–60). First, the area of positive reactions was calculated; then, the percentage of total positive reaction in sections was calculated based on the area of sections. Vacuole size measurement and positive immunoreaction are shown in Figure 1.

### 2.6. Statistical Analysis

Statistical analysis was performed using Graphpad Prism 5 software (San Diego, CA, USA, ver. 5.01). Significance level was $\alpha = 0.05$ for all statistical tests. One-way ANOVA was performed followed by Tukey's multiple comparison tests to compare distinct data sets as multiple group analysis. For some cases, an unpaired *t*-test was performed to compare distinct data sets (SREBP-1). For the comparison of UCP-1 expression and vacuole size, Kruskal–Wallis test, Dunn's multiple comparison post-test and unpaired *T*-tests were applied as post-tests. For the statistical analysis of body weight, the weekly change was calculated as percent of initial body mass; then, a two-way ANOVA test (considering the two factors: time and treatment) was performed. By running the Bonferroni's post-test, the influence of time/treatment on body weight change was evaluated. The dose-dependence of the effect was tested by comparing the two doses of the same drug at distinct time points. Values declared in the results section are mean ± SEM (for normal distribution) and median [range] (for non-normal distribution data sets).

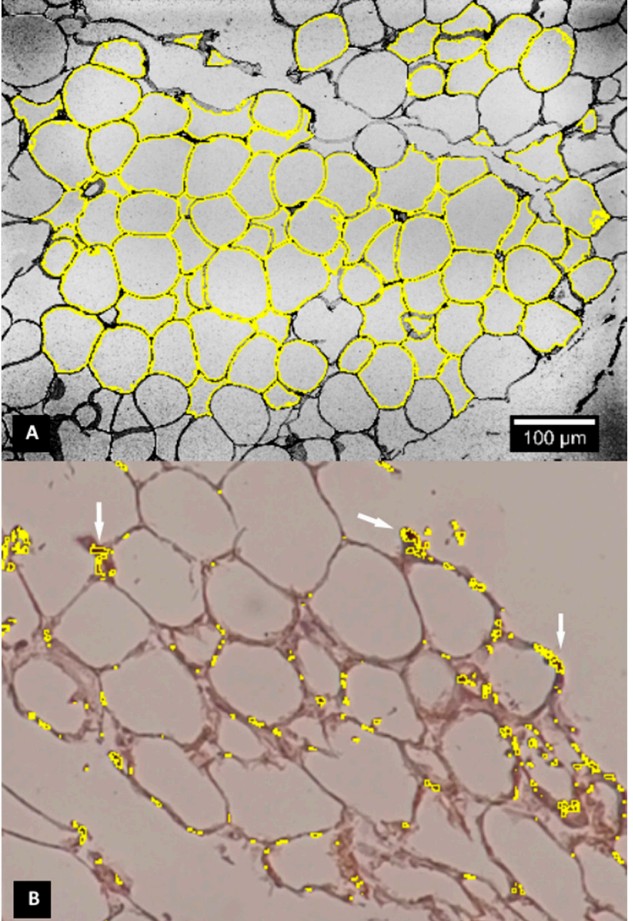

**Figure 1.** Representation of vacuole size measurement and SREBP-1 and UCP-1 expression determination using ImageJ. (**A**) The vacuole size measurement. Images were first transformed in 8-bit black and white, then color inversion and particle analysis were performed (circularity 0–1, size 100–10,000 $\mu m^2$). The vacuoles are outlined with yellow. (**B**) Quantification of positive reaction from section of adipose tissue. First, the total area of adipose tissue in section was measured, then the areas corresponding to the positive reaction were selected (having RGB code R: 40–120, G: 20–60, B: 20–60—outlined with yellow) and measured. Larger spots of positive reaction are shown with arrows.

## 3. Results

### 3.1. Body Weight

At the beginning of the experiment the animals weighted 262.38 ± 18.10 g (mean ± SD) and presented mainly a steady weight gain throughout the first weeks, then a minor fall was observed for the majority of groups. The treatment with Ola increased slightly the body weight change, that was not statistically significant (Figure 2A). From the fifth week, this pattern was disrupted, and a fall was seen. Both Ari and Car exerted weight-gain reducing effects. In the case of Ari a significant difference between the two treated groups was observed from the third week of treatment. Ari 3 mg/kg displayed statistically significant differences compared to the Ctr. Administration of the higher dose had a pronounced effect, which indicates dose-dependence (Figure 2B). A striking weight-gain reducing effect was found following the administration of Car (Figure 2C). More than that, at the fifth week of the treatment this weight change was dose-dependent. At the sixth week, the body mass differences between animals treated with the two different doses of Car were tempered. The results of the statistical analysis are shown in Table 2, graphical representation of the weight changes in Figure 2.

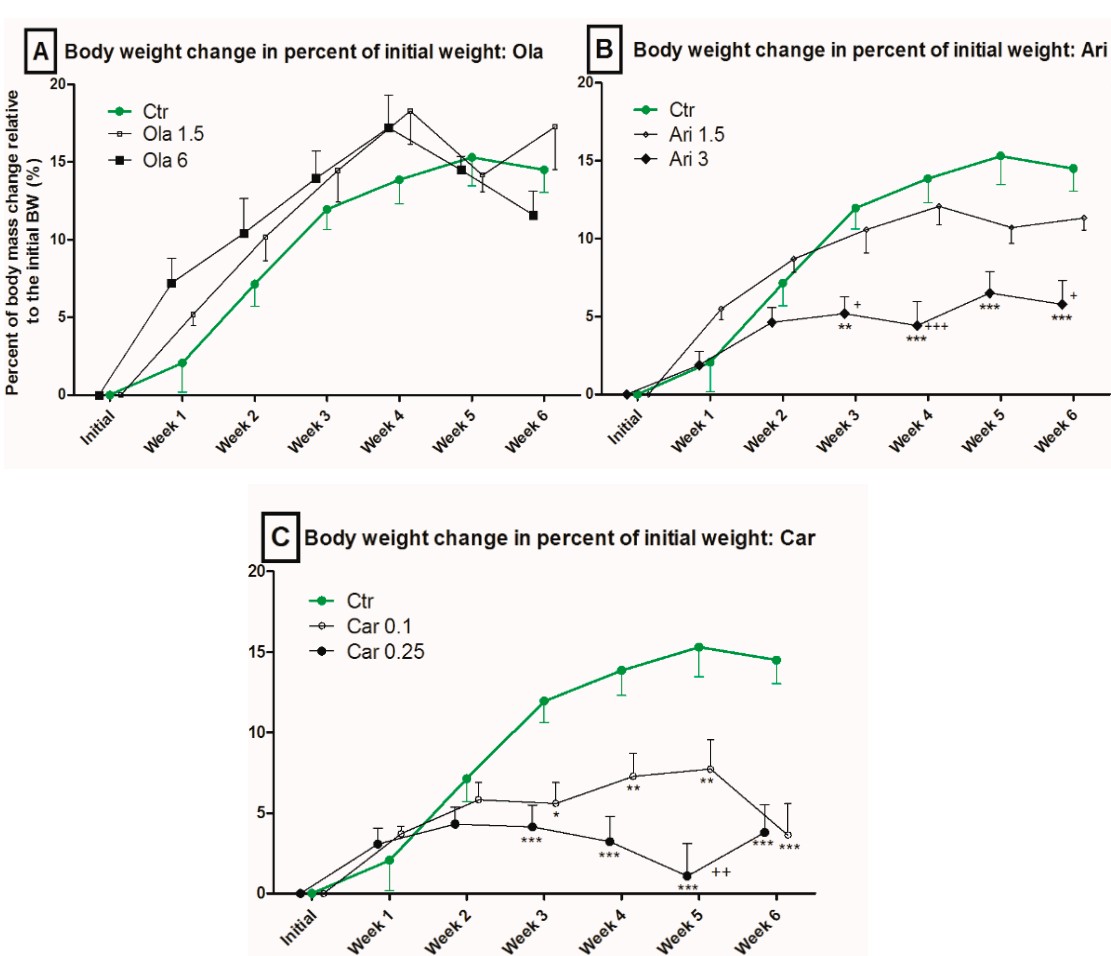

**Figure 2.** The body weight change (gain/loss) during the drug administration period expressed as percent relative to the initial weight. (**A**) Groups treated with Ola and Ctr. (**B**) Groups treated with Ari and Ctr. (**C**) Groups treated with Car and Ctr. Values displayed are means with SEM. Statistically significant differences (compared to the Ctr of the same week) are noted with * ($p < 0.05$), ** ($p < 0.01$), *** ($p < 0.001$). Significant differences between treatment regimens with two doses of the same drug are noted with + ($p < 0.05$), ++ ($p < 0.01$), +++ ($p < 0.001$).

**Table 2.** Results of two-way ANOVA analysis of the body weight change expressed as a percent of initial body mass.

| Drug | Statistical Analysis | Source of Variation | | |
|------|----------------------|---------------------|---|---|
| | | **Treatment** | **Time** | **Interaction** |
| **Ola** | FDf, Residual = F | $F_{2,84} = 0.96$ | $F_{6,84} = 71.84$ | $F_{12,84} = 1.85$ |
| | *p* value summary | ns. ($p = 0.403$) | *** ($p < 0.0001$) | ns. ($p = 0.053$) |
| Ari | FDf, Residual = F | $F_{2,90} = 8.84$ | $F_{6,90} = 75.13$ | $F_{12,90} = 7.24$ |
| | *p* value summary | ** ($p = 0.0029$) | *** ($p < 0.0001$) | *** ($p < 0.0001$) |
| Car | FDf, Residual = F | $F_{2,90} = 10.88$ | $F_{6,90} = 25.43$ | $F_{12,90} = 9.31$ |
| | *p* value summary | ** ($p = 0.0012$) | *** ($p < 0.0001$) | *** ($p < 0.0001$) |

### 3.2. Vacuole Size

The control groups vacuole size found in this experiment is similar to the ones reported recently by Verma and coworkers [31]. The adipocyte's vacuole size was significantly influenced by the treatment, as evidenced by the Kruskal–Wallis test ($p < 0.0001$). The Dunn's multiple comparison test result showed that, with the exception of Ola 6 mg/kg, all of the applied treatment regimens increased the vacuole sizes significantly. Marked rise in vacuole areas can be observed in Ola 1.5 mg/kg, Ari 1.5 mg/kg,

Ari 3 mg/kg and Car 0.25 mg/kg treatment groups compared to Ctr (Ola 1.5 mg/kg: 1246 [141.9, 11390], Ari 1.5 mg/kg: 2398 [205.4, 7703], Ari 3 mg/kg: 2129 [215.3, 8005] and Car 0.25 mg/kg 2079 [115, 8834] vs. Ctr 1354 [49.87, 9917] $\mu m^2$). A slight, yet significant increase can be seen after Car 0.1 mg/kg treatment compared to Ctr (Car 0.1 mg/kg: 1600 [101, 7818] $\mu m^2$) (Figure 3).

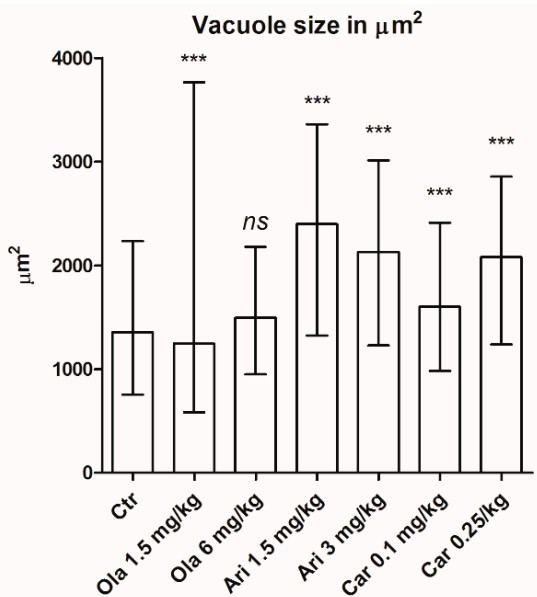

**Figure 3.** Adipocyte vacuole size expressed in $\mu m^2$ (data presented as median with interquartile range) of different groups. Statistically significant differences compared to the Ctr (Kruskal–Wallis test and Dunn's multiple post-hoc comparison) are noted with *** ($p < 0.001$). Great dispersion of vacuole sizes can be observed, because large lipid-laden vacuoles, as well as preadiocytes were present in great number, thus resulted considerable deviation of data.

### 3.3. Adipose Tissue SREBP-1 and UCP-1 Expression

Ola did not influence the expression of SREBP-1 (Ola 1.5 mg/kg: $0.28 \pm 0.04\%$; Ola 6 mg/kg: $0.31 \pm 0.05\%$ vs. Ctr: $0.27 \pm 0.08\%$) (Figure 4A) nor that of UCP-1 (Ola 1.5 mg/kg: $1.6 \pm 0.30\%$; Ola 6 mg/kg: $1.25 \pm 0.05\%$ vs. Ctr: $1.34 \pm 0.09\%$) (Figure 4B).

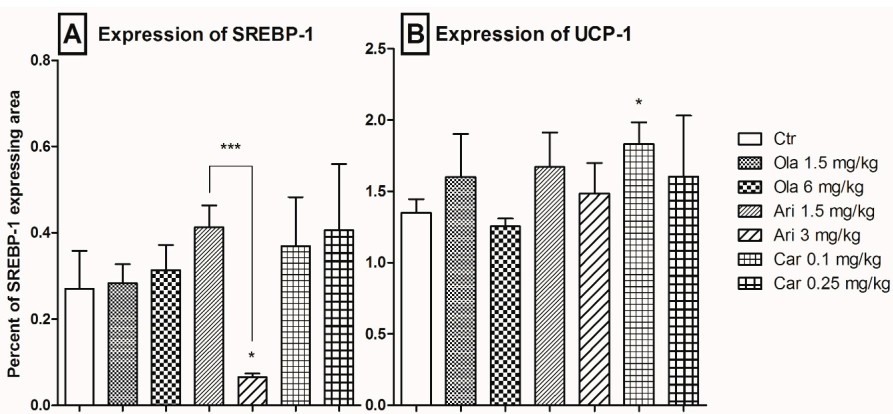

**Figure 4.** Adipose tissue protein levels of SREBP-1 (**A**) and UCP-1 (**B**). Values represent the percent of SREBP-1 expressing area in sections (quantified as positive reaction). A striking decrease of SREBP-1 can be observed in case of the high dose Ari treatment compared to Ctr, and there is a highly significant difference between the two doses of Ari (**A**). A significant increase in UCP-1 expression can be observed after low-dose Car treatment (**B**). Significant differences are noted with * ($p < 0.05$) and *** ($p < 0.001$).

Ari in lower dose displayed no considerable effect while lowered dramatically the SREBP-1 protein levels in higher dose (Ari 1.5 mg/kg: 0.41 ± 0.05%; Ari 3 mg/kg: 0.06 ± 0.01%) (Figure 4A). A slight, yet insignificant increase can be observed in terms of UCP-1 protein levels after both doses of Ari (Ari 1.5 mg/kg: 1.67 ± 0.24%; Ari 3 mg/kg: 1.48 ± 0.21%) (Figure 4B).

Car did not influence the SREBP-1 expression (Car 0.1 mg/kg: 0.36 ± 0.11%; Car 0.25 mg/kg: 0.40 ± 0.15%; Ctr: 0.27 ± 0.08%) (Figure 4A). The lower dose significantly increased the expression of UCP-1, while the greater dose did not influence it (Car 0.1 mg/kg: 1.83 ± 0.15%; Car 0.25 mg/kg: 1.41% [0.735, 3.61]; Ctr: 1.349 ± 0.09%) (Figure 4B).

## 4. Discussion

In the present study, the metabolic effects of three SGAs were tested in a head-to-head design study in rats. In this way, we compared Car, a recently marketed drug to SGAs with well-known mAE, placing it on the virtual scale ranging from Ari to Ola.

### 4.1. Effect on Body Weight

Contrary to the well-known tendency of Ola in human clinical setting, inducing a marked weight gain, in this animal model of chronic SGA administration a mild, insignificant body mass increase was observed from the first week of drug administration (Figure 2A). After four weeks of steady increase, a marked fall was observed on the fifth week for both groups treated with Ola (Figure 2A). This tendency continued for the Ola 6 mg/kg group even in the last week of treatment (Figure 2A). This phenomenon (paradox decrease after an initial weight-gain phase) has already been described after high dose Ola treatment (Ola 6 mg/kg group) in a previous experiment [28]. In the mentioned paper, the authors observed different signs of toxicity in the treated group [28]. Nevertheless, the binding profile of Ola could help to understand the dyslipidemias caused by this drug. Largely accepted, the Ola's relatively strong affinity towards $H_1$ receptors is a plausible explanation for weight gain [7,32]. The blockade of these receptors in hypothalamus results enhanced appetite [33,34]. This is also supported by a series of experimental results from knock-out animal studies [35,36]. Likewise, the 5-$HT_{2A/2C/1A}$ receptors are also implicated in the weight gain and glucose dysregulation seen with this drug [7]. Concerning the dyslipidemia, the role of PPARs have long been hypothesized [7]. In order to assess the involvement of AT-related mechanisms, we previously measured under in vitro conditions the expression of PPAR-γ after SGA administration during adipogenesis [37]. Although the protein expression (at protein level—measured by Western blot) was not changed under in vitro conditions in the mentioned experience, in 3T3-L1 cell line others found an increase in PPAR-γ expression by Ola [17]. It is worth emphasizing that the administration of PPAR-γ agonists in various in vivo experiments reverses the mAE of SGAs [38–40]. Other strategies that counteract the mAE, including betahistine administration and various herbal extracts also involve of PPAR-related mechanisms [41–43]. Taken all together, it seems probable, that PPAR is a main contributor to the mentioned phenomena. Reducing the locomotor activity of the animals and reducing the energy expenditure offers another possible explanation for the weight gain caused. This hypothesis is in accordance with the clinical experience because Ola is known to cause somnolence/sedation [44].

Ari significantly tempered the weight gain beginning with the third week of administration in a dose-dependent fashion (Figure 2B). Activation of PPAR expression and/or target genes of PPAR could be a potential mechanism behind this phenomenon. We previously studied the three SGA's effect on PPAR expression, and found no significant effect under in vitro conditions by this drug [37]. Nevertheless, the effect of Ari on body weight found in this experiment are mirroring the human clinical features of the drug, being also in accordance with the most of the previous animal experimental data. As mentioned before, low $H_1$ and 5-$HT_{2C}$ affinities are potential explanation for this. Under in vivo conditions such as in this experiment, the behavioral effects of the drug might also help to understand this effect. In this respect, Ari can be both activating and sedating under certain circumstances [44]. Regardless, further experiments are urged in order to elucidate this feature of the drug.

Car influenced the body weight in a specific manner: body weight gain was markedly attenuated beginning with the third week of drug administration. This observation is in accordance with the binding profile of Car, that is a partial agonist at $5HT_{1A}$, and low affinity for $H_1$ receptors. As mentioned earlier, another possible explanation for the lipid dysregulations might be the modulation of PPARs. Our previous in vitro results in this respect showed a significant decrease of PPAR-$\gamma$ expression at the protein level after 21 days of treatment with Car in a mouse fibroblast model of adipogenesis [37]. Considering the recently described activating effect of Car, increasing the physical activity might offer explanation to some extent, for the mAE of this drug [5,44–46].

### 4.2. Adipose Tissue-Specific Effects

The effects of Ola on SREBP-1 and UCP-1 were minor, expression was not influenced significantly (some increase of UCP-1 can be observed in the lower dose). Vacuole size was increased only in the Ola 1.5 mg/kg group. Taken all together, we can state, that in this experiment, the studied adipose-tissue specific effects of Ola are less pronounced. Previously, Skrede et al. described a significant increase of SREBP-1 mRNA in subcutaneous white AT but not in parametrial (visceral) AT and an increase of SREBP-1 protein level in subcutaneous AT [18]. Under in vitro conditions others found a similar increase of this factor after Ola exposure [17,43]. By stimulating this pro-adipogenic factor, Ola is capable of increasing the adiposity, thus can potentially induce adiposopathy. Indeed, Skrede and coworkers found a significant increase in mesenteric, parametrial, and total adipose tissue weight in Ola-treated rats [18]. In the mentioned experiment, the authors assessed the adipokines leptin and adiponectin. The levels of these were slightly increased in the Ola treated animals (statistically insignificant). This suggests that the increase in adiposity is one of the initial events that eventually lead to adiposopathy/metabolic syndrome. In order to test this hypothesis, longer experiments would be ideal, where adipose-tissue specific alterations in time could be observed parallel to mAE. The UCP-1 expression was not altered, meaning that the drug does not promote the browning of AT. Another early event is the up-regulation of several factors and subsequent down-regulation of SREBP-1 and SREBP-cleavage activating protein in the liver, resulting enhanced lipo- and cholesterol-genesis [47]. Others even found a fall in UCP-1 content of brown AT hence, the inhibition of brown adipose tissue formation seems to be a hallmark of Ola-induced metabolic derangements [48].

Both doses of Ari caused an increase in adipocyte vacuole size. It is worth to mention, that we have previously observed a similar phenomenon after fluoxetine treatment, that similarly decreases body weight but concomitantly increases the adipocyte vacuole size in visceral adipose tissue [49,50]. Ari in high dose dramatically lowered the SREBP-1 expression. This taken together with the increased vacuole size (reflecting a pro-adipogenic effect) means that Ari is enhancing adipogenesis, without stimulating SREBP-1 expression (possibly by influencing other pro-adipogenic factors). Other teams reported no significant effect on SREBP-1 expression in visceral AT [47,51]. Nonetheless, in the present experiment, we found a marked lowering in SREBP-1 expression in higher dose. Further studies are needed in order to elucidate the significance of this finding. Considering UCP-1 expression, Ari caused an increasing tendency that might explain to some extent the weight gain tempering effect of the drug.

Car did not influence significantly the expression of SREBP-1 (Figure 4A). Given that SREBP-1 is implicated in de novo lipo- and adipo-genesis, this finding suggests that Car is not causing profound changes in the function of adipose tissue (at least not by means of SREBP-1). This is consistent with the weight gain moderation seen with this drug (Figure 2C). Car significantly increased the UCP-1 expression in visceral AT (Figure 4B). This represents a beneficial mechanism that can lead to browning of the AT, increasing the oxidative capacity of the organ, conferring protection against the most common mAE of this drug. Indeed, overexpression of UCP-1 under in vitro conditions increases the glucose uptake of adipocytes, thereby it is hypothesized to be a mechanism that could counteract insulin resistance, transforming the white adipocyte into an efficient glucose sink [52]. Accordingly, under in vivo conditions, the activation of brown AT or browning of white AT increases the expression of UCP-1, therefore, results in weight loss via increased energy expenditure. This process also lowers

plasma TG and glucose in various animal models (and also confers protection against atherosclerosis in a transgenic mouse model) [53–56]. Moreover, Kooijman and coworkers have shown, that activation of brown adipose tissue or browning of white AT decreases lipid droplet content (i.e., vacuole size) in mice [53]. In this experiment, Car's effect on UCP-1 expression (Figure 4B) is in accordance with the weight gain moderating effect (Figure 2C). However, careful interpretation of this finding is advised considering the major differences in brown adipose tissue biology between humans and rodents. First, rodents have a very high weight-adjusted metabolic rate, that couples to a higher body surface/mass ratio and also higher whole-body thermal conductance compared to humans [57]. Consequently, rodents have considerably larger brown adipose tissue depots than humans do (1.5 vs. 0.2% of whole-body weight, respectively) [57]. Further, thermogenic signaling also differs between humans and rodents (ex. β3 adrenergic receptors dramatically increase the thermogenesis in rodents, but the effects are milder in human) [58]. Taken all together, one can observe that, in rodents, thermogenesis has a much more important role in energy metabolism than in humans.

Considering the increased vacuole size caused by this drug, similar to that seen in Ari treated groups, it seems plausible, that Car is causing a mild, apparently healthy expansion of AT. This result conflicts with the decrease in lipid droplet content after brown AT activation/browning described by Kooijman et al. [53]. It must be underlined that, based on this experiment, the existence of complementary mechanisms that may come into play cannot be ruled out. A possible system that might come into play in this regard is the endocannabinoid system. As the components of this system are expressed both at peripherical and central levels, it governs metabolic processes including lipo- and adipogenesis. As demonstrated earlier by Lazzari and coworkers, the effects of Ola administration are mediated by the endocannabinoid system both centrally and at the level of the adipose tissue [59]. Similar mechanisms might be responsible for the adipose tissue effects of Ari and Car in the present experiment. Future experiments that study the involvement of the endocannabinoid system in these effects after Ari and Car administration are also urged.

Furthermore, the increase of the vacuole size of adipocytes from the visceral adipose tissue can mean an increased tendency of preadipocytes to accumulate TG and undergo adipogenesis, but may also indicate a decreased cell number in AT accumulating the same amount of TGs. This can be triggered either by inducing early growth arrest of preadipocytes or by limiting the mitotic clonal expansion (the first two stages of adipogenesis) [11]. We cannot exclude the latter scenarios, since the expression of early differentiating factors that indicate the engagement of cells in differentiation (a stage when the number of cells reach the plateau) were not measured. To the best of our knowledge, this is the first paper reporting the SREBP-1 and UCP-1 expression after chronic Car administration.

### 4.3. Limitations

Limitations of the study include technical aspects: the technique used for estimating protein expression (immunohistochemistry) differs from the gold standard in molecular biology (Western blot). At the same time, the parameters studied were determined at the treatment end-point of the experiment (after 6 weeks of drug self-administration), reflecting only a momentary, temporary status. It is well known that the effects of same drug (e.g., the influence on the body weight) can be very different according to the period of administration.

Another important limitation of the extrapolation of these findings to humans is the major difference in brown adipose tissue biology and its role in energy metabolism.

### 5. Conclusions

Car is a new SGA with favorable effect on body weight (similar to Ari, inducing a considerable, dose-dependent reduction in body weight gain). We conclude that Car is situated closer to the Ari's end of the scale of SGAs, with a favorable effect on body weight. The adipose tissue-related mechanisms only partially explain these aspects.

**Author Contributions:** Conceptualization, L.-I.B., Z.G., M.K. and I.-Z.K.; investigation, L.-I.B., Z.P., Z.V.V., B.K. and B.H.; resources, Z.G., M.K., Z.P., Z.V.V. and B.K.; writing—original draft preparation, L.-I.B.; writing—review and editing, Z.G., M.K. and I.-Z.K.; visualization, L.-I.B.; supervision, I.-Z.K.; project administration, L.-I.B. and I.-Z.K.; funding acquisition, L.-I.B., M.K. and Z.G. All authors have read and agreed to the published version of the manuscript.

**Funding:** This research was funded by the Transylvanian Museum Society and Semmelweis University (grant nr. 84.5/2017).

**Acknowledgments:** LIB was supported by Collegium Talentum 2019 Programme of Hungary.

**Conflicts of Interest:** The funders had no role in the design of the study; in the collection, analyses, or interpretation of data; in the writing of the manuscript, or in the decision to publish the results.

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
