# Peer review of "Effect on Body Weight and Adipose Tissue by Cariprazine: A Head-to-Head Comparison Study to Olanzapine and Aripiprazole in Rats"

_scipharm, doi:10.3390/scipharm88040050_

Round 1

Reviewer 1 Report

Authors tried to explore the weight-gain reducing effect of Cariprazine in comparison to olanzapine and aripiprazole in rats.

Title of the manuscript shows the metabolic profile of Cariprazine, but the metabolic profile information was not included in the manuscript. Serum lipid profiling data in rats can be included in the manuscript for better explanation of weight-gain reducing effect and lipolysis process. Title of the manuscript should be changed based on the overall results.

This study does not show the images of vacuole size of adipocytes and immunohistochemistry results. These data should be included in the manuscript.

Line 4- rat should be plural.

Line 28- Need to define the Car.

Line 34- Need to indicate the abbreviation of Cariprazine.

Line 38- Authors need to provide a valid reason for the inclusion of prolactin in the keywords section.

Line 50- Replace satiety- with safety-

Line 53- Need to define the Car.

Line 53- need to mention the reference for the statement.

Line 88- SREBP1 not SPREBP1

Line 109- in vitro and in vivo words should be in italics throughout the manuscript.

Line 116- What is the purity of Car? Purity of Car should be stated.

Line 116- City name should be mentioned for the Sanghai Resuperpharma Tech ltd.

Line 127- Number of animals used for the study should be included.

Line 127- Age of the animals should be included before administering these drugs.

Line 127- Acclimatization period of animals should be stated before the treatment.

Line 127- Authors need to include the reference for the duration of treatment.

Line 147- What is the dose and composition of Ketamine-Xylazine used in anesthesia of animals? What is the source of ketamine and xylazine? Authors are required to include this information and cite the reference.

Line 148- Required to include the centrifugation speed and time information.

Line 168- Authors said that liver tissues were processed in immunohistochemistry section. Authors should include the results of liver protein levels of SREP1 and UCP1 and effect of these drugs. 

Line 221- Error bars are very high in figure 2. Authors need to provide the possible reason for this. Authors should check the figure 2 data calculations and statistical analysis data.

Line 245- rat should be plural.

Author Response

Response to Reviewer 1 Comments

Title of the manuscript shows the metabolic profile of Cariprazine, but the metabolic profile information was not included in the manuscript. Serum lipid profiling data in rats can be included in the manuscript for better explanation of weight-gain reducing effect and lipolysis process. Title of the manuscript should be changed based on the overall results.

Autor response: The lipid disturbances caused by second-generation antipsychotics are har to mirror in the animal experimental setting, and thus we decided not to include them in the observed experimental parameters. The title was changed accordingly.

This study does not show the images of vacuole size of adipocytes and immunohistochemistry results. These data should be included in the manuscript.

Autor response: Figure 1 was inserted, showing (A) vacuole size measurement and (B) positive immunoreaction quantification (Lines 180-187).

Line 4- rat should be plural.

Autor response: Changed accordingly.

Line 28- Need to define the Car.

Autor response: Definition made before abbreviation, in the abstract.

Line 34- Need to indicate the abbreviation of Cariprazine.

Autor response: Cariprazine was defined earlier (line 4), hence, the acronym was used.

Line 38- Authors need to provide a valid reason for the inclusion of prolactin in the keywords section.

Autor response: Since the keyword `prolactin` was used by mistake (earlier, unsubmitted manuscript versions contained the prolactin elevation, as an adverse effect of SGA treatment). Since corresponding paragraphs were not included in the final, uploaded manuscript, prolactin was removed from the keyword list.

Line 50- Replace satiety- with safety-

Autor response: the section was removed.

Line 53- Need to define the Car.

Autor response: section removed.

Line 53- need to mention the reference for the statement.

Autor response: Reference inserted: Citrome L. The ABC’s of dopamine receptor partial agonists - Aripiprazole, brexpiprazole and cariprazine: The 15-min challenge to sort these agents out. Int J Clin Pract. 2015;69(11):1211–20.

Line 88- SREBP1 not SPREBP1

Autor response: Corrected accordingly

Line 109- in vitro and in vivo words should be in italics throughout the manuscript.

Autor response: These expressions were formatted accordingly throughout the manuscript.

Line 116- What is the purity of Car? Purity of Car should be stated.

Autor response: The purity statement was made in manuscript (line 119).

Line 116- City name should be mentioned for the Sanghai Resuperpharma Tech ltd.

Autor response: The city name is mentioned.

Line 127- Number of animals used for the study should be included.

Autor response: The animal number used (per treatment group) in our study was stated in the Materials and Methods section (line 128). Total number of animals was also inserted.

Line 127- Age of the animals should be included before administering these drugs.

Autor response:  Age of animals mentioned in line 128.

Line 127- Acclimatization period of animals should be stated before the treatment.

Autor response:   Acclimatization period mentioned in lines 139-140.

Line 127- Authors need to include the reference for the duration of treatment.

Autor response:  The duration of administration was chosen based on previous reports and also our team`s experience in experimental work with rats, information and references included (lines 143-145).

Line 147- What is the dose and composition of Ketamine-Xylazine used in anesthesia of animals? What is the source of ketamine and xylazine? Authors are required to include this information and cite the reference.

Autor response:  Anesthesia details (doses, source of anesthetics, references) were defined in text (line 149).

Line 148- Required to include the centrifugation speed and time information

Autor response:  Sentence was removed from manuscript.

Line 168- Authors said that liver tissues were processed in immunohistochemistry section. Authors should include the results of liver protein levels of SREP1 and UCP1 and effect of these drugs.

Autor response:  The expression of these factors were not measured in the mentioned tissues. These were performed, in order to test the robustness of our IHC system as a whole. First, the positive reaction needs to be evidenced, by preparing a tissue sample (under the same conditions as the samples of interest) that is known to express the target protein (liver for SREBP1 and brown adipose tissue for UCP-1). The second part of this paragraph details the procedure, when we tested the false positive reaction (i.e. nonspecific binding of the secondary antibody).

Line 221- Error bars are very high in figure 2. Authors need to provide the possible reason for this. Authors should check the figure 2 data calculations and statistical analysis data.

Autor response:  The data was drawn from a considerably heterogenous population in which small values (preadipocytes with reduced vacuolar area) and large vacuoles were both in great number. This explanation was inserted in the caption of fig. 2.

Line 245- rat should be plural.

Autor response:  Corrected accordingly, line 252

Reviewer 2 Report

Reviewer’s report:

Title:  Metabolic profile of cariprazine: a head-to-head comparison study to olanzapine and aripiprazole in rat

Authors: László-István Bába, Zsolt Gáll, Melinda Kolcsár, Zsuzsánna Pap, Zoltán Varga, Béla Kovács, Beatrix Hack, Imre-Zoltán Kun

Adverse metabolic effects constitute a severe limitation in the use of antipsychotic drugs. Therefore, each new drug in this group must be assessed in terms of its effect on the body weight and metabolic profile. In their work, László-István Bába et al. compared the metabolic effects of cariprazine to olanzapine and aripiprazole, finding that the administration of the new drug decreases weight in rats probably due to the increase in thermogenic activity of adipose tissue. In my opinion, the study constitutes a significant contribution to our understanding of cariprazine action. Therefore, I have only some minor remarks that should be clarified before the manuscript is published.

Minor revisions:

Whole text

The full name of the substance “cariprazine” occurs in the title and the abstract, while in the manuscript itself, the acronym "Car" is used only. Technically, the acronym should be introduced at the beginning of the Introduction (as in the case of olanzapine (Ola) and  aripiprazole (Ari)) and then used consistently throughout the text.

Abstract

Lines 30-31 “The drugs were self-administered in two different doses by female, adult, Wistar rats for six weeks." – I would suggest changing to "self-administered in two different doses to female, adult, Wistar rats."

Introduction

Although this part of the work is intended to introduce the reader to the subject of the work and to present the purpose of the research, in its current form, it is a bit too extensive. I would suggest shortening it a bit so that it contains content directly related to the research topic and finish it with a clear presentation of the study's aims.

Lines 101-102 “UCP-1 is responsible for the transport of protons through the internal membrane of 101 the mitochondria resulting eventually heat instead of ATP”- please consider adding the preposition “in” – “resulting eventually in”

Material and Methods

Lines 146-148: “At the end of the six weeks treatment period, the animals were sacrificed by ketamine-xylazine anesthesia and cardiac punctures were performed. Blood samples were taken and immediately centrifuged, serum samples were kept on -20 °C until further processing.” - What was the purpose of serum collection? As I understand for future projects because, in this work, I did not find the measurements carried out in the serum.

Discussion

This part of the manuscript is relatively well balanced. The only aspect missing is deliberation whether the thermogenic effects of cariprazine observed in rats may be relevant in humans. It is worth mentioning because, in rodents, thermogenesis plays a more significant role in regulating the body's energy balance than in humans.

Author Response

Response to Reviewer 2 Comments

Whole text

The full name of the substance “cariprazine” occurs in the title and the abstract, while in the manuscript itself, the acronym "Car" is used only. Technically, the acronym should be introduced at the beginning of the Introduction (as in the case of olanzapine (Ola) and aripiprazole (Ari)) and then used consistently throughout the text.

Autor response: Changed accordingly.

Abstract

Lines 30-31 “The drugs were self-administered in two different doses by female, adult, Wistar rats for six weeks." – I would suggest changing to "self-administered in two different doses to female, adult, Wistar rats."

Autor response:  Changed, accordingly.

Introduction

Although this part of the work is intended to introduce the reader to the subject of the work and to present the purpose of the research, in its current form, it is a bit too extensive. I would suggest shortening it a bit so that it contains content directly related to the research topic and finish it with a clear presentation of the study's aims.

Autor response:  Some parts were deleted, others shortened, in order to keep this section concise and sound.

Lines 101-102 “UCP-1 is responsible for the transport of protons through the internal membrane of 101 the mitochondria resulting eventually heat instead of ATP”- please consider adding the preposition “in” – “resulting eventually in”

Autor response:  Added, accordingly.

Material and Methods

Lines 146-148: “At the end of the six weeks treatment period, the animals were sacrificed by ketamine-xylazine anesthesia and cardiac punctures were performed. Blood samples were taken and immediately centrifuged, serum samples were kept on -20 °C until further processing.” - What was the purpose of serum collection? As I understand for future projects because, in this work, I did not find the measurements carried out in the serum.

Autor response:  Indeed, the collection of blood samples has no direct impact on our current experiment and was removed.

Discussion

This part of the manuscript is relatively well balanced. The only aspect missing is deliberation whether the thermogenic effects of cariprazine observed in rats may be relevant in humans. It is worth mentioning because, in rodents, thermogenesis plays a more significant role in regulating the body's energy balance than in humans.

Autor response:  These considerations were added to the discussion (362-371) and limitation sections (399-400).

Reviewer 3 Report

Baba and colleagues present a manuscript on the characterization of some side effects of second-generation antipsychotics such as olanzapine and others. In particular, the authors compare the effects of cariprazine on energy metabolism with those of olanzapine and aripiprazole. In addition, the authors make an interesting molecular study on adipose tissue. The manuscript is clear and linear, and partially, it contributes to the identification of new therapeutic strategies.

However, some critical issues require clarification by the authors.

In detail:

1) The authors base the effects of the three drugs on body weight only. However, since the drug is administered through food, it is necessary to know, in quantitative terms, how much the animals eat (kcal and grams of food consumed). Indeed, weight loss can be linked to low food consumption. Furthermore, it would also be interesting to evaluate how much food the animals consume on average before the experiment begins. Finally, it is necessary to know how much water the animals have drunk. For all these reasons, I wonder if the animals were housed in single cages or lived in communities.

2) The authors should better explain the "vacuole size" experiments. Authors should also consider the weight of the entire white adipose tissue. In this regard, see: Lazzari P, Serra V, Marcello S, Pira M, Mastinu A. Metabolic side effects induced by olanzapine treatment are neutralized by CB1 receptor antagonist compounds co-administration in female rats. Eur Neuropsychopharmacol. 2017; 27 (7): 667-678. doi: 10.1016 / j.euroneuro.2017.03.010

3) The authors should better represent figure 2. It would be preferable to use symbols for each sample of the experimental group.

4) From figure 1 it is difficult to understand the different effects on body weight of the 3 drugs. Could they use just one chart for everything?

5) Aripiprazole and cariprazine appear to reduce body weight. How do the authors explain this effect? Which receptor systems are involved? In this regard, see: Lazzari P, Serra V, Marcello S, Pira M, Mastinu A. Metabolic side effects induced by olanzapine treatment are neutralized by CB1 receptor antagonist compounds co-administration in female rats. Eur Neuropsychopharmacol. 2017; 27 (7): 667-678. doi: 10.1016 / j.euroneuro.2017.03.010

6) Finally, the authors should perform some behavioral tests to highlight the antipsychotic effect of cariprazine at the dosages used. Furthermore, they do not perform any pharmacokinetic or pharmacodynamic measures that could support their work.

Author Response

Response for Reviewer 3 Comments

Baba and colleagues present a manuscript on the characterization of some side effects of second-generation antipsychotics such as olanzapine and others. In particular, the authors compare the effects of cariprazine on energy metabolism with those of olanzapine and aripiprazole. In addition, the authors make an interesting molecular study on adipose tissue. The manuscript is clear and linear, and partially, it contributes to the identification of new therapeutic strategies.

Autor response:  thank you for the positive feed-back.

However, some critical issues require clarification by the authors.

 In detail:

1) The authors base the effects of the three drugs on body weight only. However, since the drug is administered through food, it is necessary to know, in quantitative terms, how much the animals eat (kcal and grams of food consumed). Indeed, weight loss can be linked to low food consumption. Furthermore, it would also be interesting to evaluate how much food the animals consume on average before the experiment begins. Finally, it is necessary to know how much water the animals have drunk. For all these reasons, I wonder if the animals were housed in single cages or lived in communities.

Autor response:  The most important problem with the mentioned experimental procedure is that this needs to be performed in so-called metabolic cages, where chow and water consumption could have robustly been measured. In our experimental facility, we had simple polycarbonate cages, suitable for single housing, and did not intend to evaluate the food consumption as these measurements could not have been made exactly. We confirm that animals were housed separately, because the drug self-administration was not possible under pair or group-housed conditions.

2) The authors should better explain the "vacuole size" experiments. Authors should also consider the weight of the entire white adipose tissue. In this regard, see: Lazzari P, Serra V, Marcello S, Pira M, Mastinu A. Metabolic side effects induced by olanzapine treatment are neutralized by CB1 receptor antagonist compounds co-administration in female rats. Eur Neuropsychopharmacol. 2017; 27 (7): 667-678. doi: 10.1016 / j.euroneuro.2017.03.010

Autor response: The adipose tissue was not weighted because we focused on molecular/transcriptional changes rather than gross quantity of the tissue.

3) The authors should better represent figure 2. It would be preferable to use symbols for each sample of the experimental group.

Autor response:  Evaluation of the effect on vacuole size was made by measuring a high number of cell vacuole surface. These values were added into separate columns, thus the values compared are distinct cell vacuole areas by treatment group. Since that large number of values cannot be represented on a graph, we decided to plot the interquartile and medians instead (distribution was non-Gaussian).

4) From figure 1 it is difficult to understand the different effects on body weight of the 3 drugs. Could they use just one chart for everything?

Autor response:  Figure 1 was compacted as requested.

5) Aripiprazole and cariprazine appear to reduce body weight. How do the authors explain this effect? Which receptor systems are involved? In this regard, see: Lazzari P, Serra V, Marcello S, Pira M, Mastinu A. Metabolic side effects induced by olanzapine treatment are neutralized by CB1 receptor antagonist compounds co-administration in female rats. Eur Neuropsychopharmacol. 2017; 27 (7): 667-678. doi: 10.1016 / j.euroneuro.2017.03.010

Autor response:  Explanation regarding CB1 receptors included in discussion section (lines 377-383).

6) Finally, the authors should perform some behavioral tests to highlight the antipsychotic effect of cariprazine at the dosages used. Furthermore, they do not perform any pharmacokinetic or pharmacodynamic measures that could support their work.

Autor response:

Antipsychotic effect was not assessed, since this effect of the drug is well documented and robustly reproduced by different animal models. Doses of Car used in this study correspond to the ones used by Watson and coworkers in their study demonstrating the antipsychotic effect of Car [1].

  1. Watson DJG, King M V., Gyertyán I, et al. The dopamine D3-preferring D2/D3 dopamine receptor partial agonist, cariprazine, reverses behavioural changes in a rat neurodevelopmental model for schizophrenia. Eur Neuropsychopharmacol. 2016;26:208–224.

Round 2

Reviewer 3 Report

The authors have improved the unclear aspects of the manuscript, in my opinion, now the manuscript could be published.